# Crosstalk in Facial EMG and Its Reduction Using ICA

**DOI:** 10.3390/s23052720

**Published:** 2023-03-02

**Authors:** Wataru Sato, Takanori Kochiyama

**Affiliations:** 1Psychological Process Research Team, Guardian Robot Project, RIKEN, 2-2-2 Hikaridai, Seika-cho, Soraku-gun, Kyoto 619-0288, Japan; 2Field Science Education and Research Center, Kyoto University, Oiwake-cho, Kitashirakawa, Kyoto 606-8502, Japan; 3Brain Activity Imaging Center, ATR-Promotions, 2-2-2 Hikaridai, Seika-cho, Soraku-gun, Kyoto 619-0288, Japan

**Keywords:** corrugator supercilii, crosstalk, facial electromyography (EMG), independent component analysis (ICA), zygomatic major

## Abstract

There is ample evidence that electromyography (EMG) signals from the corrugator supercilii and zygomatic major muscles can provide valuable information for the assessment of subjective emotional experiences. Although previous research suggested that facial EMG data could be affected by crosstalk from adjacent facial muscles, it remains unproven whether such crosstalk occurs and, if so, how it can be reduced. To investigate this, we instructed participants (*n* = 29) to perform the facial actions of frowning, smiling, chewing, and speaking, in isolation and combination. During these actions, we measured facial EMG signals from the corrugator supercilii, zygomatic major, masseter, and suprahyoid muscles. We performed an independent component analysis (ICA) of the EMG data and removed crosstalk components. Speaking and chewing induced EMG activity in the masseter and suprahyoid muscles, as well as the zygomatic major muscle. The ICA-reconstructed EMG signals reduced the effects of speaking and chewing on zygomatic major activity, compared with the original signals. These data suggest that: (1) mouth actions could induce crosstalk in zygomatic major EMG signals, and (2) ICA can reduce the effects of such crosstalk.

## 1. Introduction

There is extensive evidence from psychophysiological studies that facial electromyography (EMG) can provide valuable information for the assessment of subjective emotional experience [1,2,3]. Specifically, EMG signals recorded from the corrugator supercilii muscle (related to frowning) and zygomatic major muscle (related to smiling) are negatively and positively associated with subjective valence ratings, respectively. For example, a previous study has recorded continuous subjective ratings of valence and EMG from these muscles during the observation of emotional films [4]. The dynamic changes in subjective valence ratings were negatively and positively associated with corrugator supercilii and zygomatic major EMG activity, respectively. Although there remain debates regarding the universal relationships between emotional categorical states and facial muscle activation patterns [5,6,7], ample evidence suggests that emotional valence is reliably related to facial muscle activity [8]. Although some studies have shown that facial EMG responses occur even without the subjective experiences of emotional events [9,10], suggesting possible dissociation between subjective and physiological emotional responses, ample evidence suggests that they are generally coordinated and constitute a unified emotional system [11]. Several recent studies have used facial EMG of the corrugator supercilii and zygomaticus major muscles for emotion sensing during various active tasks, including conversation [12,13] and food consumption [14,15].

Some investigators have cautioned that the use of facial EMG signals as a proxy for the emotional state may be affected by crosstalk [16,17]; that is, the EMG signal for a specific muscle may be affected by electric activity in adjacent muscles through volume conduction. Indeed, crosstalk is a serious concern in all types of surface EMG data [18,19], but particularly for facial EMG recordings because crosstalk in surface EMG is distance-dependent [20] and there are more than two dozen individual muscles in proximity on each side of the face [21].

However, there remains uncertainty regarding whether and how crosstalk affects facial EMG during tasks related to emotion sensing. Only a few studies have empirically investigated this issue, and the results have been equivocal [22,23,24]. In one study, facial EMG signals were recorded from seven muscles in the lower half of the face, including the zygomatic major muscle [22]. Participants performed six tasks that required the production of discrete facial actions. Although a certain degree of crosstalk was present, it was within acceptable limits for most muscles. However, the study did not record corrugator EMG signals or statistically analyze zygomatic major EMG data; it also failed to assess facial actions in the context of emotional tasks. Another study recorded facial EMG from five muscles, including the corrugator supercilii and zygomatic major, while participants performed six simple and combined facial actions (e.g., smiling and chewing) [23]. The researchers concluded that the degree of crosstalk was acceptably small for all facial actions, except for a large effect of chewing on zygomatic major activity. However, the conclusion was not supported by statistical tests. Data from another study of 29 facial actions using 48 monopolar electrodes suggested that crosstalk did not fundamentally change facial EMG patterns, possibly because EMG amplitudes decrease according to the square of the distance from the source [24]. Based on these findings, we hypothesized that mouth actions, such as eating and talking, would produce a small but meaningful degree of crosstalk in EMG signals from the zygomatic major muscle, which is close to the mouth, but not in signals from the corrugator supercilii muscle.

In addition, uncertainty remains regarding whether data analysis techniques can reduce the effect of crosstalk on facial EMG. A potentially useful technique is independent component analysis (ICA), which performs blind source separation via unsupervised learning [25] and can decompose electrophysiological sensor signals into independent components (ICs) that correspond to source activities [26]. Several studies have shown that ICA on electroencephalography data can perform linear spatial filtering on the recorded data to the effects of summing the volume-conducted cortical source activities in each recording channel [26]. Because surface EMG signals are the sum of the propagating action potentials produced by the recruited motor units (i.e., the basic muscle-building block consisting of one motor neuron and all the muscle fibers that it innervates) [27], ICA may effectively decompose motor unit source activities. Consistent with this notion, some studies have demonstrated that ICA effectively decomposed target muscle signals and crosstalk within surface EMG signals recorded from hand muscles [28,29]. Although ICA was also applied to facial EMG signals, the results were equivocal [30,31,32,33]. In a seminal work [30], EMG signals were recorded from hand and face muscles, including the zygomatic major, while participants performed hand gestures and uttered vowels, respectively. The EMG signals were then decomposed using ICA, and the gestures/vowels were classified through artificial neural network analysis of the ICs. The classification accuracies of the gestures and vowels were 100% and ~60%, respectively; the researchers concluded that ICA performed poorly with respect to the classification of facial EMG. Other studies used ICA to evaluate facial EMG signals recorded from a few muscles, including the zygomatic major [31,32]. The researchers reported that their artificial neural network analysis of ICs accurately classified smiling. Another study recorded facial EMG signals using an array of electrodes on the cheeks; the results showed that ICA of the EMG signals successfully distinguished ICs related to three different facial actions, including smiling [33]. Based on these data, although there is no direct evidence regarding the reduction of crosstalk in facial EMG, we hypothesized that ICA could be used to remove crosstalk from facial EMG signals related to emotion-sensing tasks.

To test these hypotheses, we measured facial EMG while participants performed facial actions. We measured EMG from the corrugator supercilii and zygomatic major muscles, as well as the masseter and suprahyoid muscles, which are involved in mouth actions such as chewing and talking. We instructed the participants to perform frowning, smiling, chewing, and speaking actions, in isolation and combination. Initially, we focused on EMG signals from emotion-sensing-related muscles during simple facial actions of non-target muscles (e.g., zygomatic major EMG activity during speaking) to identify the presence of crosstalk. We also evaluated whether crosstalk could emerge during combined facial actions. Because facial actions are generally difficult to perform consciously [22], we did not expect to observe fully isolated muscle contractions. Subsequently, we used ICA to determine whether crosstalk could be reduced.

## 2. Materials and Methods

### 2.1. Participants

Twenty-nine Japanese volunteers (16 women; mean ± standard deviation age, 22.6 ± 2.7 years) participated in this study. The sample size was determined through *a priori* power analysis conducted using G*Power software ver. 3.1.9.2 [34]. We assumed paired *t*-tests (two-tailed) to compare the original and ICA-reconstructed signals, to detect the mean effect size in biological psychological studies (i.e., *d* = 0.8 [35]) with an *α*-level of 0.05 and power (1–*β*) of 0.80. Power analysis showed that >15 participants were required. All participants had normal or corrected-to-normal visual acuity. After an explanation of the experimental procedure, all participants provided written informed consent. This study was approved by the Ethics Committee of the Unit for Advanced Studies of the Human Mind, Kyoto University (approval number: 30-P-6). All experiments adhered to the ethical policies of our institution and the Declaration of Helsinki.

### 2.2. Apparatus

The experiments were performed using Presentation software (Neurobehavioral Systems, Berkeley, CA, USA) and a Windows computer (HP Z200 SFF, Hewlett-Packard Japan, Tokyo, Japan). Slides showing the task instructions were presented on a 19-inch cathode ray tube monitor (HM903D-A; Iiyama, Tokyo, Japan) with a resolution of 1024 × 768 pixels.

### 2.3. Procedure

The experiments were carried out in a soundproof, electrically shielded chamber (Science Cabin, Takahashi Kensetsu, Tokyo, Japan). Following electrode attachment, participants were instructed to perform facial actions while their facial EMG signals were recorded. All facial actions (i.e., frowning, smiling, chewing, speaking [i.e., slowly uttering vowel sounds], frowning + chewing, smiling + chewing, frowning + speaking, and smiling + speaking) were listed on the screen. To facilitate understanding, the screen also showed pictures depicting the anatomy of the facial muscles, along with photographs of single actions. Participants were asked to practice all facial actions at their own pace. Then, after 8 practice trials, a total of 64 experimental trials were performed. The order of conditions was pseudorandomized.

For each trial, after the action instructions (e.g., “Frown”) had been presented for 3 s, followed by the presentation of a small black cross for 3 s as a fixation point, a large red cross appeared on the screen for 5 s. Participants were asked to perform facial actions in accordance with instructions provided during the presentation of the red cross.

### 2.4. EMG Recording

EMG signals were recorded from the corrugator supercilii, zygomatic major, masseter, and suprahyoid muscles on the left side of the face (Figure 1). Sets of pre-gelled, self-adhesive 0.7-cm Ag/AgCl electrodes (1.5-cm interelectrode spacing; Prokidai, Sagara, Japan) were used. The electrodes were placed in accordance with guidelines and methods used in previous studies [24,36,37,38]. A ground electrode was placed on the middle of the forehead. The data were amplified, bandpass-filtered (20–400 Hz), and sampled at 1000 Hz using an EMG-025 amplifier (Harada Electronic Industry, Sapporo, Japan), the PowerLab 16/35 data acquisition system with a 16-bit A/D resolution and LabChart Pro 8.0 software (ADInstruments, Dunedin, New Zealand). A low-cut filter (20 Hz) was used to remove motion artifacts [39]. Participants’ behaviors were monitored by video, which was unobtrusively recorded using a digital web camera (HD1080P; Logicool, Tokyo, Japan).

### 2.5. Data Analysis

Data preprocessing and ICA were performed using Psychophysiological Analysis Software 3.3 (Computational Neuroscience Laboratory of the Salk Institute, La Jolla, CA, USA) and in-house programs implemented in the MATLAB R2021a environment (MathWorks, Natick, MA, USA). Preprocessing was conducted in a manner identical to that used in a previous study [14]. EMG data for each trial were recorded at baseline (beginning 500 ms before stimulus onset) and during stimulus presentation (during the performance of facial actions; 5000 ms). Then, the data were rectified and downsampled to 10 Hz; this resampling was conducted because ICA assumes zero-lag synchronization [40], and the conductance velocity of crosstalk in surface EMG is reportedly ~4.5 m/s [41]. EMG preprocessing using rectification and downsampling was recommended in guidelines [42] and used in several previous studies (e.g., [43]).

The processed EMG data for each trial and participant were then concatenated and subjected to ICA, which enables blind source separation of a linear mixture of sources in electrophysiological signals that are spatially fixed and temporally independent [26]. We used the infomax algorithm [25,44], which identifies the unmixing matrix by maximizing the joint entropy (i.e., maximizing the individual entropies while minimizing the mutual information) of the resulting unmixed signals. The artificial neural network was trained using unmixing weighted matrices that maximized the joint entropy of transformed channel data [26]. For all participants, ICA identified the most important ICs for single muscles. Then, to remove crosstalk associated with the masseter and suprahyoid muscles, EMG signals were reconstructed using the two ICs that exhibited the highest variance with respect to the corrugator supercilii and zygomatic major muscle EMG data. Figure 2 and Appendix A present representative examples of original and ICA-reconstructed EMG signals. The original and ICA-reconstructed EMG signals were baseline-corrected with respect to the mean value over the pre-stimulus period and averaged across the stimulus presentation period (5000 ms).

Statistical tests were conducted using JASP 0.14.1 software [45]. Original EMG signals were tested for the differences from zero using one-sample *t*-tests (two-tailed). Subsequently, original and ICA-reconstructed EMG signals were compared using paired *t*-tests (two-tailed). All results were considered statistically significant at *p* < 0.05 after correction for multiple tests (i.e., eight) for each measure using Holm’s sequential method [46]. Cohen’s *d* values [47] were reported as the effect size measures. Our preliminary analysis indicated that several measures had a non-normal distribution (Shapiro-Wilk test, *p* < 0.05). Although *t*-tests should be considered asymptotically valid under general conditions, even when the normality assumption is rejected [48], we additionally conducted non-parametric Wilcoxon signed-rank tests to confirm the results of the *t*-tests.

## 3. Results

### 3.1. Original EMG Signal Analysis

Figure 3 shows the mean ± standard error values for original and ICA-reconstructed EMG signals.

First, using one-sample *t*-tests, original EMG signals of the corrugator supercilii and zygomatic major muscles were evaluated to determine whether the isolated non-target facial actions also elicited muscle activation (i.e., crosstalk). The results (Table 1; Appendix A) showed that the corrugator supercilii was significantly activated only during frowning (*t*(28) = 5.42, *p* < 0.001, *d* = 1.01). The zygomatic major muscle was significantly activated during the target smiling action, as well as non-target frowning, chewing, and speaking (*t*(28) > 2.26, *p* < 0.032, *d* > 0.41).

Next, corrugator supercilii and zygomatic major EMG activities during combined facial actions were analyzed using the methods described above. The corrugator supercilii was significantly activated only during the frowning + speaking and frowning + chewing action combinations (*t*(28) > 3.73, *p* < 0.001, *d* > 0.68). The zygomatic major showed significant EMG activity during all combined actions, including the conditions only activating non-target muscles (i.e., the frowning + speaking and frowning + chewing actions) (*t*(28) > 2.40, *p* < 0.024, *d* > 0.44).

To evaluate the validity of the methods used to manipulate facial actions, EMG signals recorded from the masseter and suprahyoid muscles during both simple and combined actions were analyzed. The results revealed significant EMG activity in all conditions except frowning alone (*t*(28) > 3.90, *p* < 0.001, *d* > 0.72).

To confirm the robustness of these results, original EMG signals were analyzed using one-sample Wilcoxon signed-rank tests, which confirmed the significant results of one-sample *t*-tests (Appendix A).

### 3.2. Comparison of Original and ICA-Reconstructed EMG Signals

To evaluate the ability of ICA to reduce crosstalk (i.e., masseter and suprahyoid muscle activities) from the EMG signals of the corrugator supercilii and zygomatic major muscles, original and ICA-reconstructed signals were compared using paired *t*-tests.

First, corrugator supercilii and zygomatic major EMG signals during simple action conditions were evaluated. The results (Table 2; Appendix A) showed no significant differences in corrugator supercilii activity (*p* > 0.05, Holm-corrected). For zygomatic major activity, significant differences were found during all actions, indicating that the ICA-reconstructed EMG signals were weaker than the original signals (*t*(28) > 2.61, *p* < 0.015, *d* > 0.48).

Next, EMG signals recorded during combined actions were evaluated using the methods described above. For corrugator supercilii activity, the ICA-reconstructed signals were significantly stronger than the original signals during frowning + chewing (*t*(28) = 2.97, *p* = 0.006, *d* = 0.55). For zygomatic major activity, significant differences were found during all combined actions, indicating weaker ICA-reconstructed signals than the original signals (*t*(28) > 3.26, *p* < 0.004, *d* > 0.60).

To confirm the validity of reducing masseter and suprahyoid muscle activity using ICA, EMG signals recorded from those muscles during both simple and combined actions were also analyzed. For both muscles, the original signals were stronger than the ICA-reconstructed signals under all conditions except frowning alone (*t*(28) > 2.71, *p* < 0.012, *d* > 0.50).

The original and ICA-reconstructed signals were compared using non-parametric paired Wilcoxon signed-rank tests. The results showed that all significant effects according to *t*-tests were also significant on Wilcoxon signed-rank tests, except for corrugator supercilii activity during frowning + chewing (Appendix A).

## 4. Discussion

Our original EMG signal analysis confirmed that deliberate facial actions appropriately activated all four target muscles. Importantly, crosstalk arising from the contraction of other muscles during frowning, speaking, and chewing affected the zygomatic major EMG signals. These results are consistent with the previous suggestion that crosstalk is present among facial EMG data obtained in emotion-sensing paradigms [16]. We observed crosstalk for zygomatic major activity, but this was less evident for corrugator supercilii activity. This is consistent with a previous study regarding the effect of chewing [23], although that study did not include a thorough statistical analysis. Our results are also compatible with the suggestion that crosstalk in EMG is distance-dependent [20]. Extending prior research, the present study provides reliable evidence that crosstalk arising from non-emotional facial actions can affect facial EMG signals recorded during emotion sensing.

Furthermore, our results of the comparison of original and ICA-reconstructed EMG signals demonstrated that ICA reduced the effect of crosstalk arising from mouth actions on zygomatic major EMG signals. This finding corroborates the results of studies in which ICA effectively removed crosstalk from hand muscle EMG signals [28,29] and distinguished facial EMG signals [31,32,33]. However, in another study where ICA was used to evaluate facial EMG data, ICs exhibited poor vowel classification performance [30]. This equivocal result may be explained by methodological differences: the study with poor performance included only one participant, and thus may have lacked sufficient power to reveal the effects of ICA. Our method allowed us to detect a statistically significant effect of ICA on facial EMG signals. To the best of our knowledge, this study provides the first evidence that ICA can reduce crosstalk in facial EMG signals recorded for the purpose of emotion sensing.

The present results have several practical implications. Emotions have a major impact on happiness [49], behavior, and health [50]; however, self-report measures of these aspects are inherently subjective, subject to biases, and difficult to record in a continuous manner during tasks [51]. Therefore, emotion sensing on the basis of physiological signals (i.e., facial EMG) is advantageous [1,2,3]. Further, as some recent studies have developed wearable devices that can record facial EMG signals [33,52,53], future studies presumably will use facial EMG to detect emotional states under naturalistic situations. However, our results suggest that non-emotional facial actions (e.g., speaking and eating) can affect emotion-related EMG. At the same time, they also indicate that ICA can reduce the effect of crosstalk. We hope that our findings will enhance the sensitivity of future analyses of emotion sensing. 

Some limitations of the present study should be acknowledged. First, we only tested two types of non-emotional facial actions; crosstalk arising from other actions requires investigation. We may have failed to detect crosstalk in corrugator supercilii EMG signals because we only tested mouth actions. Facial actions in the upper face region, such as eyebrow-raising [54], should be tested. Furthermore, facial EMG has applications beyond emotion sensing, including human–computer interface [55,56], oral processing and food texture analysis [57,58,59], speech and swallowing disorder assessment [60,61,62], and facial palsy assessment [63,64], which may have a specific target and confounding facial muscle activities. Future research should explore additional facial actions and their effect on crosstalk. Second, our comparative analysis of the original and ICA-reconstructed corrugator supercilii EMG signals unexpectedly revealed higher values for the ICA-reconstructed data, suggesting that crosstalk removal improves EMG signals under specific conditions. However, these results may be related to artifacts associated with the statistical manipulation; further studies are needed to evaluate such effects of ICA.

## 5. Conclusions

In this study, speaking and chewing induced EMG activity in the zygomatic major muscle. Compared with the original signals, the ICA-reconstructed zygomatic major EMG signals were less affected by speaking and chewing. These data indicate that mouth actions can induce crosstalk in zygomatic major EMG signals; importantly, ICA can reduce the effects of such crosstalk. However, because we tested only a limited number of facial actions and our results showed some unexpected patterns, further studies are warranted to investigate crosstalk in facial EMG and its ICA analysis.

## Figures and Tables

**Figure 1 sensors-23-02720-f001:**
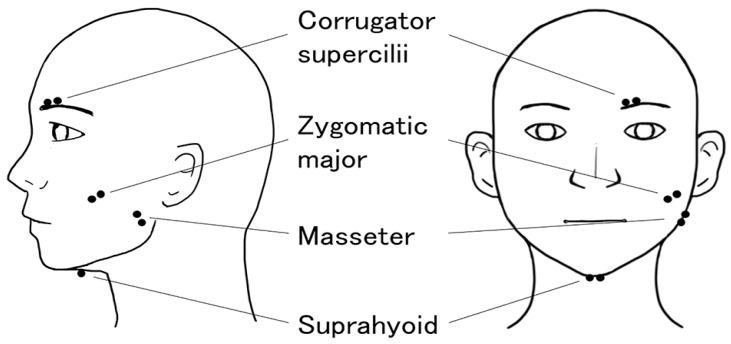
Schematic illustrations of electrode placement for the recording of electromyography signals from the corrugator supercilii, zygomatic major, masseter, and suprahyoid muscles.

**Figure 2 sensors-23-02720-f002:**
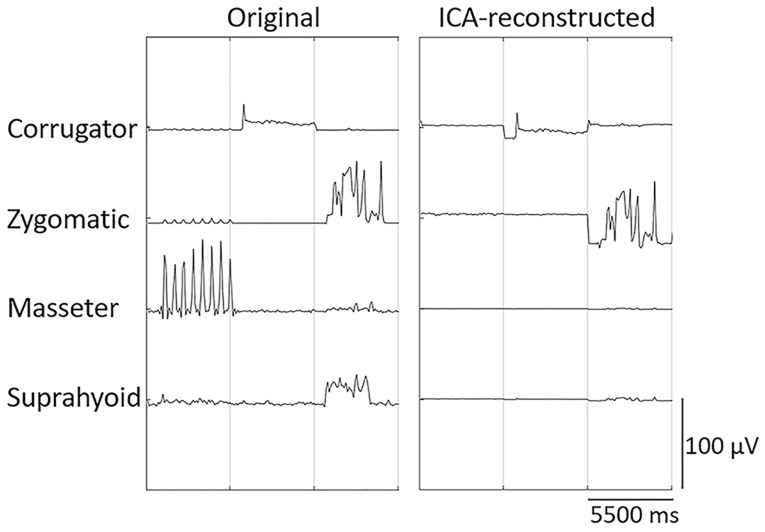
Examples of original and independent component analysis (ICA)-reconstructed electromyography signals from the corrugator supercilii, zygomatic major, masseter, and suprahyoid muscles. Data for three trials (a total of 16.5 s; each trial contained 0.5-s pre- and 5-s post-stimulus periods) of a representative participant are shown. To remove crosstalk arising from the masseter and suprahyoid muscle activities, the signals were reconstructed using the independent components that exhibited the highest variance with respect to the corrugator supercilii and zygomatic major muscle activities.

**Figure 3 sensors-23-02720-f003:**
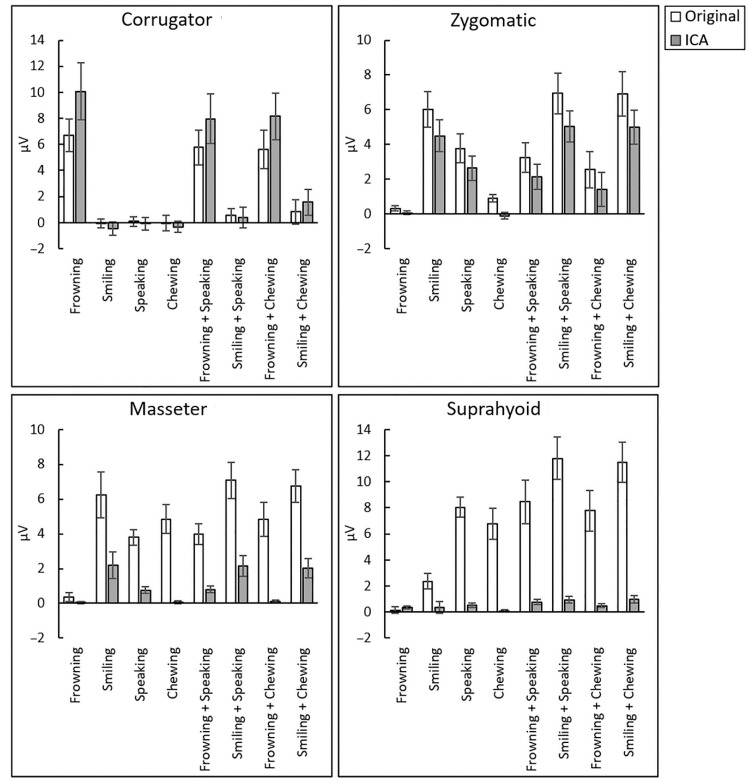
Mean ± standard error of original and independent component analysis (ICA)-reconstructed electromyography data recorded from the corrugator supercilii, zygomatic major, masseter, and suprahyoid muscles during frowning, smiling, chewing, speaking actions, and combinations of these actions.

**Table 1 sensors-23-02720-t001:** Results of one-sample *t*-tests (vs. zero; two-tailed) of original electromyography signals.

Muscle	Statistic	Facial Action							
		Frowning	Smiling	Speaking	Chewing	Frowning + Speaking	Smiling + Speaking	Frowning + Chewing	Smiling + Chewing
Corrugator	*t*	**5.42**	0.13	0.24	0.08	**4.36**	0.95	**3.74**	0.87
	*p*	**<0.001**	0.895	0.809	0.939	**<0.001**	0.349	**<0.001**	0.39
	*d*	**1.01**	0.03	0.05	0.01	**0.81**	0.18	**0.69**	0.16
Zygomatic	*t*	**2.27**	**5.88**	**4.47**	**4.23**	**3.86**	**5.88**	**2.41**	**5.40**
	*p*	**0.031**	**<0.001**	**<0.001**	**<0.001**	**<0.001**	**<0.001**	**0.023**	**<0.001**
	*d*	**0.42**	**1.09**	**0.83**	**0.79**	**0.72**	**1.09**	**0.45**	**1.00**
Masseter	*t*	1.52	**4.74**	**8.45**	**5.98**	**6.75**	**6.84**	**4.89**	**7.28**
	*p*	0.140	**<0.001**	**<0.001**	**<0.001**	**<0.001**	**<0.001**	**<0.001**	**<0.001**
	*d*	0.28	**0.88**	**1.57**	**1.11**	**1.25**	**1.27**	**0.91**	**1.35**
Suprahyoid	*t*	0.59	**3.91**	**10.44**	**5.69**	**5.07**	**7.30**	**4.91**	**7.49**
	*p*	0.558	**<0.001**	**<0.001**	**<0.001**	**<0.001**	**<0.001**	**<0.001**	**<0.001**
	*d*	0.11	**0.73**	**1.94**	**1.06**	**0.94**	**1.36**	**0.91**	**1.39**

*d*, Cohen’s *d* statistic [47]. Degrees of freedom were 28 for all tests. Significant results (*p* < 0.05) corrected using Holm’s method are shown in bold font.

**Table 2 sensors-23-02720-t002:** Results of paired *t*-tests (two-tailed) comparing original and independent component analysis-reconstructed signals.

Muscle	Statistic	Facial Action							
		Frowning	Smiling	Speaking	Chewing	Frowning + Speaking	Smiling + Speaking	Frowning + Chewing	Smiling + Chewing
Corrugator	*t*	2.39	1.97	0.52	1.26	2.28	0.40	**2.97**	1.00
	*p*	0.024	0.059	0.606	0.218	0.031	0.691	**0.006**	0.325
	*d*	0.44	0.37	0.10	0.23	0.42	0.08	**0.55**	0.19
Zygomatic	*t*	**2.62**	**3.10**	**3.78**	**6.15**	**4.58**	**3.27**	**6.01**	**3.57**
	*p*	**0.014**	**0.004**	**<0.001**	**<0.001**	**<0.001**	**0.003**	**<0.001**	**0.001**
	*d*	**0.49**	**0.58**	**0.70**	**1.14**	**0.85**	**0.61**	**1.12**	**0.66**
Masseter	*t*	1.40	**2.95**	**6.63**	**5.87**	**6.06**	**5.03**	**4.77**	**4.95**
	*p*	0.173	**0.006**	**<0.001**	**<0.001**	**<0.001**	**<0.001**	**<0.001**	**<0.001**
	*d*	0.26	**0.55**	**1.23**	**1.09**	**1.13**	**0.93**	**0.89**	**0.92**
Suprahyoid	*t*	0.77	**2.72**	**9.40**	**5.71**	**4.72**	**6.50**	**4.69**	**6.58**
	*p*	0.450	**0.011**	**<0.001**	**<0.001**	**<0.001**	**<0.001**	**<0.001**	**<0.001**
	*d*	0.14	**0.51**	**1.75**	**1.06**	**0.88**	**1.21**	**0.87**	**1.22**

*d*, Cohen’s *d* statistic [47]. Degrees of freedom were 28 for all tests. Significant results (*p* < 0.05) corrected using Holm’s method are shown in bold font.

## Data Availability

The data supporting the findings of this study are available within the Appendix A.

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
