# Peer review of "Crosstalk in Facial EMG and Its Reduction Using ICA"

_sensors, 2023, doi:10.3390/s23052720_

Round 1

Reviewer 1 Report

Report on ‘Crosstalk in facial EMG and its reduction using ICA’ by Wataru Sato and Takanori Kochiyama.

This work addresses the analysis of the EMG signals registered from the facial muscles, particularly the issues of crosstalk from adjacent facial muscles. The Authors test the independent component analysis (ICA) to reduce the effects of such crosstalk. The neural network was trained to maximize the entropy of transformed channel data.

The presented statistics are convincing that the study provides evidence for ICA to reduce crosstalk in facial EMG signals. The work is well-written and relatively easy to understand. I would only like to read what kind of entropy is used in ICA, although just for the sake of precision. Further research has also been outlined, and I encourage the Authors to carry it out.

Except for a tiny request (entropy), I have no other points to raise and recommend this work publication.

Author Response

Dear Reviewer,

Thank you for your useful and constructive comments on our manuscript. We have carefully revised the manuscript according to your suggestions. All revisions were marked using the “Track Changes” function. Page numbers refer to those in the no-mark version of the manuscript. A professional English-language editing service has made language-related changes to the manuscript (http://www.textcheck.com/certificate/9HoZO3).

Point 1
I would only like to read what kind of entropy is used in ICA, although just for the sake of precision.
Response
In accordance with your suggestion, we have added the information on entropy type to the Methods section (p. 5):
“We used the infomax algorithm, which identifies the unmixing matrix by maximizing the joint entropy (i.e., maximizing the individual entropies while minimizing the mutual information) of the resulting unmixed signals.”

Sincerely yours,

Wataru Sato

Reviewer 2 Report

This is a good attempt of applying ICA for the cross talk/ noise removal from EMG signals.

Author Response

Dear Reviewer,

Thank you for your positive comments.
We have revised the manuscript according to Reviewers’ suggestions. All revisions were marked using the “Track Changes” function. Page numbers refer to those in the no-mark version of the manuscript. A professional English-language editing service has made language-related changes to the manuscript (http://www.textcheck.com/certificate/9HoZO3).

Sincerely yours, 

Wataru Sato

Reviewer 3 Report

The paper presents a study on crosstalk affecting electromyography (EMG) signals from the corrugator supercilii and zygomatic major muscles, which can be used to assess the emotional experience. 29 participants performed various facial actions such as frowning, smiling, chewing, and speaking while EMG signals were measured. The results showed that speaking and chewing activities induced crosstalk in zygomatic major EMG signals, and using independent component analysis (ICA) reduced the effects of crosstalk.

This paper may be of interest to those using EMG for expression assessment. However, several points need to be addressed:

1) Some references in the introduction are outdated. Consider using more recent literature.

2) The use of EMG for emotional assessment is controversial due to the non-universality of facial expressions. A review by Russell (1994) found that there is a cross-cultural agreement in recognition, but it is influenced by cultural and individual factors. While there is a relationship between facial expressions and emotions, more literature support is needed to affirm that EMG provides valuable information for emotional experience assessment. (Dimberg et al., 2000)

*Russell, J. A. (1994). Is there universal recognition of emotion from facial expression? A review of the cross-cultural studies.Psychological Bulletin, 115(1), 102–141. 

*Dimberg, U., Thunberg, M., & Elmehed, K. (2000). Unconscious facial reactions to emotional facial expressions. Psychological Science, 11(1), 86-89.

3) Facial EMG has many other applications besides emotion and facial expression physiology, such as speech and swallowing disorders assessment (10.1152/jappl.1999.86.5.1663), human-computer interfaces (10.1109/TNSRE.2018.2849202), neuromuscular disorders study (10.1007/s00405-020-05949-1), and oral processing and food texture analysis (10.1016/S1050-6411(02)00065-2 and 10.1016/j.physbeh.2021.113580). The introduction should provide a deeper understanding of the significance of measuring facial muscles.

4) The importance of ICA in EMG context should be better explained with a brief explanation of signal decomposition and the physiological composition of EMG signals. Consider using additional references to better support the literature review.

5) It's unclear if the data normality was tested. A specific test such as Kolmogorov-Smirnov, Shapiro-Wilk, Anderson-Darling, or Lillierfors should be used.

6) The A/D resolution of the EMG system should be provided.

7) It's unclear whether the data rectification and resampling method has been validated. Consider finding references to better support the methodological design or rerun the signal analysis using a state-of-the-art procedure. Most works calculate RMS from consecutive windows, and the resampled amplitude of the filtered and rectified signal may not contain the same information.

8) The figures need improvement:

Figure 1 should have a second face with a front view.

Figure 2 is confusing and not informative.

Figures 3 have low quality and should be exported in vector format.

Author Response

Dear Reviewer,

Thank you for your useful and constructive comments on our manuscript. We have carefully revised the manuscript according to your suggestions. All revisions were marked using the “Track Changes” function. Page numbers refer to those in the no-mark version of the manuscript. A professional English-language editing service has made language-related changes to the manuscript (http://www.textcheck.com/certificate/9HoZO3).

Point 1
Some references in the introduction are outdated. Consider using more recent literature.
Response
As suggested, we have cited additional recent studies in the Introduction (e.g., Wingenbach, 2023; pp. 1–3).

Point 2
The use of EMG for emotional assessment is controversial due to the non-universality of facial expressions. A review by Russell (1994) found that there is a cross-cultural agreement in recognition, but it is influenced by cultural and individual factors. While there is a relationship between facial expressions and emotions, more literature support is needed to affirm that EMG provides valuable information for emotional experience assessment. (Dimberg et al., 2000)
*Russell, J. A. (1994). Is there universal recognition of emotion from facial expression? A review of the cross-cultural studies.Psychological Bulletin, 115(1), 102–141.
*Dimberg, U., Thunberg, M., & Elmehed, K. (2000). Unconscious facial reactions to emotional facial expressions. Psychological Science, 11(1), 86-89.
Response
As suggested, we have discussed the controversy regarding the use of EMG for emotional assessment in the Introduction (p. 1).

Point 3
Facial EMG has many other applications besides emotion and facial expression physiology, such as speech and swallowing disorders assessment (10.1152/jappl.1999.86.5.1663), humancomputer interfaces (10.1109/TNSRE.2018.2849202), neuromuscular disorders study (10.1007/s00405-020-05949-1), and oral processing and food texture analysis (10.1016/S1050-6411(02)00065-2 and 10.1016/j.physbeh.2021.113580). The introduction should provide a deeper understanding of the significance of measuring facial muscles.
Response
As suggested, we have discussed several other applications of facial EMG (p. 10). Because the current study focused only on emotion sensing (the influence of masseter and suprahyoid muscles on the corrugator and zygomatic muscles), we have discussed this issue as a future research direction in the Discussion section.

Point 4
The importance of ICA in EMG context should be better explained with a brief explanation of signal decomposition and the physiological composition of EMG signals. Consider using additional references to better support the literature review.
Response
As suggested, we have described ICA signal decomposition and the physiological composition of EMG signals in the Introduction (p. 2).

Point 5
It's unclear if the data normality was tested. A specific test such as Kolmogorov-Smirnov, Shapiro-Wilk, Anderson-Darling, or Lillierfors should be used.
Response
As suggested, we performed Shapiro-Wilk tests to check data normality and found that several measures significantly violated the normality assumption. Therefore, we conducted Wilcoxon signed-rank tests to confirm the results of the t-tests. Almost all significant results on t-tests were confirmed to be significant using the non-parametric tests. We have presented these results briefly in the main text (pp. 6 and 9), and in detail in the Supplementary material.

Point 6
The A/D resolution of the EMG system should be provided.
Response
As suggested, we have stated that our system had a 16-bit A/D resolution in the Methods section (p. 4).

Point 7
It's unclear whether the data rectification and resampling method has been validated. Consider finding references to better support the methodological design or rerun the signal analysis using a state-of-the-art procedure. Most works calculate RMS from consecutive windows, and the resampled amplitude of the filtered and rectified signal may not contain the same information. 
Response
As suggested, we have added references supporting our data analysis methods to the Methods section (p. 4). A previous methodological article recommended EMG preprocessing conducting rectification and downsampling (Altimar et al., 2012). Additionally, several previous studies performed rectification and downsampling preprocessing for EMG data (e.g., Breteler et al., 2007).

Point 8
The figures need improvement:
Figure 1 should have a second face with a front view.
Figure 2 is confusing and not informative.
Figures 3 have low quality and should be exported in vector format.
Response
As suggested, we have depicted the front view of a second face in Figure 1; we have modified Figure 2 and improved the quality of Figure 3.

Sincerely yours, 

Wataru Sato

Round 2

Reviewer 3 Report

The effect size (d) should be explained in Material and Methods and in Table's legend. 

Significant differences should be marked in the graph (using a star over the bar graph for example). 

Include limitations and future works on Conclusions.

Author Response

Dear Reviewer,

Thank you for your patience. We have carefully revised the manuscript according to your suggestions. All revisions were marked using the “Track Changes” function.

Point 1
The effect size (d) should be explained in Material and Methods and in Table's legend.
Response
As suggested, we have explained d values in the Methods section (p. 5) and Tables’ legends (p. 8).

Point 2
Significant differences should be marked in the graph (using a star over the bar graph for example).
Response
As suggested, we have marked significant differences in the figure. Because we conducted two statistical tests (i.e., one-sample and paired t-tests), we depicted two figures in the Supplementary Material.

Point 3
Include limitations and future works on Conclusions.
Response
As suggested, we have included limitations and future works on Conclusions (p. 10).

Sincerely yours, 

Wataru Sato